# Implementation of Biosphere Reserves in Poland–Problems of the Polish Law and Nature Legacy

**Beata Raszka \*** and **Maria Hełdak**

Institute of Spatial Economy, Wrocław University of Environmental and Life Sciences, Grunwaldzka 55, 50-357 Wrocław, Poland; maria.heldak@upwr.edu.pl
* Correspondence: beata.raszka@upwr.edu.pl

**Abstract:** The article addresses the issue of the management and functioning of biosphere reserves (BRs) in Poland. The hypothesis was raised that BRs in Poland are virtual rather than real entities. The study examined how the existence of BRs is reflected in Polish strategic and planning documents. The study examined documents from 1947 to 2022, i.e., Polish legal acts (archived and current), the national Strategy for Responsible Development, voivodeship strategies, and national park protection plans. It evaluated to what extent the biosphere reserves fulfil their role in Poland, as defined by the Man and Biosphere program. To verify the research questions, legal documents (laws and regulations) enacted by the Polish authorities, strategies, and planning documents created at the central and voivodeship levels, protection plans for nature conservation forms covering biosphere reserves, economic plans of entities managing biosphere reserves, and other documents were analyzed. It was shown that: (1) BRs do not have a legal basis in Polish legislation at the national level, despite Poland's ratification of the Man and Biosphere program, (2) there is a lack of detailed information about BRs in national and voivodeship strategic documents (development strategies and spatial development plans for voivodeships), (3) the existence of biosphere reserves does not translate into spatial planning principles at the local level (municipalities), (4) there is no legal possibility to separate tasks related to biosphere reserves in nature conservation protection plans (national parks, nature reserves, and landscape parks), (5) in the case of transboundary BRs, the Inspection carried out in the Carpathians International Biosphere Reserve (Poland-Ukraine-Slovakia) showed only formal cooperation, not practical. In conclusion, the management of BRs and the implementation of tasks contained in the MaB program, particularly those related to sustainable development of the environment, society, and economy, are ineffective due to the lack of legal authorization in Poland.

**Keywords:** biosphere reserves; management; transboundary cooperation; spatial management

## 1. Introduction

In 1971, UNESCO (United Educational, Scientific and Cultural Organization) launched the international program "Man and Biosphere" [1]. The aim of the program is to achieve a sustainable balance between often conflicting goals, such as preserving biodiversity, supporting human development, and maintaining cultural values. The program initiated the creation of the World Network of Biosphere Reserves. The status of a biosphere reserve is awarded by the UNESCO World Committee to areas with outstanding natural and landscape values on a global scale, where legal forms of nature protection and sustainable economic use should create a mutually harmonized system.

In the original concept of the "Man and Biosphere" program, biosphere reserves were to be a tool for sustainable development and biodiversity conservation, allowing the harmonization of interactions between people and nature. Over the decades of the program's activity, the concept and goals of biosphere reserves have changed. Initially, the focus was on environmental protection, and the main goal was to preserve areas with undisturbed natural resources. These were to be areas in countries that include ecosystems

representative of the world's major biomes or important for various reasons for a particular country. The Statutory Framework of the World Network of Biosphere Reserves adopted at the UNESCO-MaB conference in Seville in 1995 and the newly defined strategy for creating biosphere reserves assumed that these reserves would not only be protected, isolated zones with limited access but also places where economic and socio-cultural development is carried out in balance with preserving biodiversity.

In the "Recommendations for the Establishment and Functioning of Transboundary Biosphere Reserves", adopted in Spain in 2000, a document outlining proposed actions to achieve specific goals in transboundary biosphere reserves, it was noted that specific solutions would differ depending on the location of each biosphere reserve. This is not, however, a list of mandatory actions; flexibility and adaptation to the situation in each country are essential.

During the 3rd International Congress of Biosphere Reserves of the MaB Program, the "Madrid Action Plan for 2008–2013" [2] was adopted, which aimed to develop mechanisms to support the sustainable development of biosphere reserves, among others, for effective management of climate change. At the 4th World Congress of Biosphere Reserves, the approved Action Plan for the Man and Biosphere Program and the World Network of Biosphere Reserves for 2016–2025, called the "Lima Action Plan... 2016" [3], was adopted. The actions included in it are intended to implement a new strategy adopted by the UNESCO General Conference in 2015. This strategy, which continues the Strategies from Seville (1995) [4] and Madrid (2008) [2], expands the scope of tasks related to climate change, education on sustainable development, and cooperation at the local level.

The impact of human civilization is reflected in the decline of biodiversity. It is difficult to significantly limit anthropogenic pressure, but it is possible to try to minimize its effects. Biosphere reserves, established as part of the Man and Biosphere project, were intended to be the scientific response to civilizational challenges. As part of the environmental debate, ref. [5] showed that "the Anthropocene philosophy narrative transforms MAB's philosophy but sustainable development continues to steer its guidance, revealing a gap between philosophy and practice as the Anthropocene is institutionalized". These units, encompassing areas of natural value but not devoid of anthropogenic influences, were meant to serve as laboratories for analyzing the coexistence of environment, society, and economy. In many countries, existing biosphere reserves are an example of sustainable development recognized by local communities. The aim of this review article is to assess to what extent international commitments and conventions regarding the Man and Biosphere program are being realized in Poland through national legislation, economic and social development strategies, and implementing documents (spatial planning and nature conservation plans and programs).

## 2. Materials and Methods

The text discusses a research study conducted in Poland on the implementation of the MaB program in the country's biosphere reserves. Currently, there are 11 biosphere reserves established in Poland; 5 transboundary and 6 located exclusively on Polish territory (Figure 1, Table 1). Due to their potential importance for local sustainable development, especially legally formalized cross-border cooperation: Poland–Ukraine–Slovakia, Poland–Belarus, Poland–Bohemia, it was decided to examine to what extent biosphere reserves in Poland fulfill the MaB program assumptions. The following research questions were posed:

1.  Is a biosphere reserve in Poland a functional structure noted in Polish legal, planning, and strategic documents?
2.  Is a biosphere reserve in Poland a tool for nature protection?
3.  Does the existence of a biosphere reserve have an impact on local development by supporting social initiatives?
4.  Are promotional and educational activities and scientific research carried out in biosphere reserves, in accordance with local and national sustainable development

programs? If so, what are they about, are they long-term and systematic, or ad hoc and individual?

5. Is cross-border cooperation sufficient? Are there legal and financial mechanisms to support such cooperation in Poland?

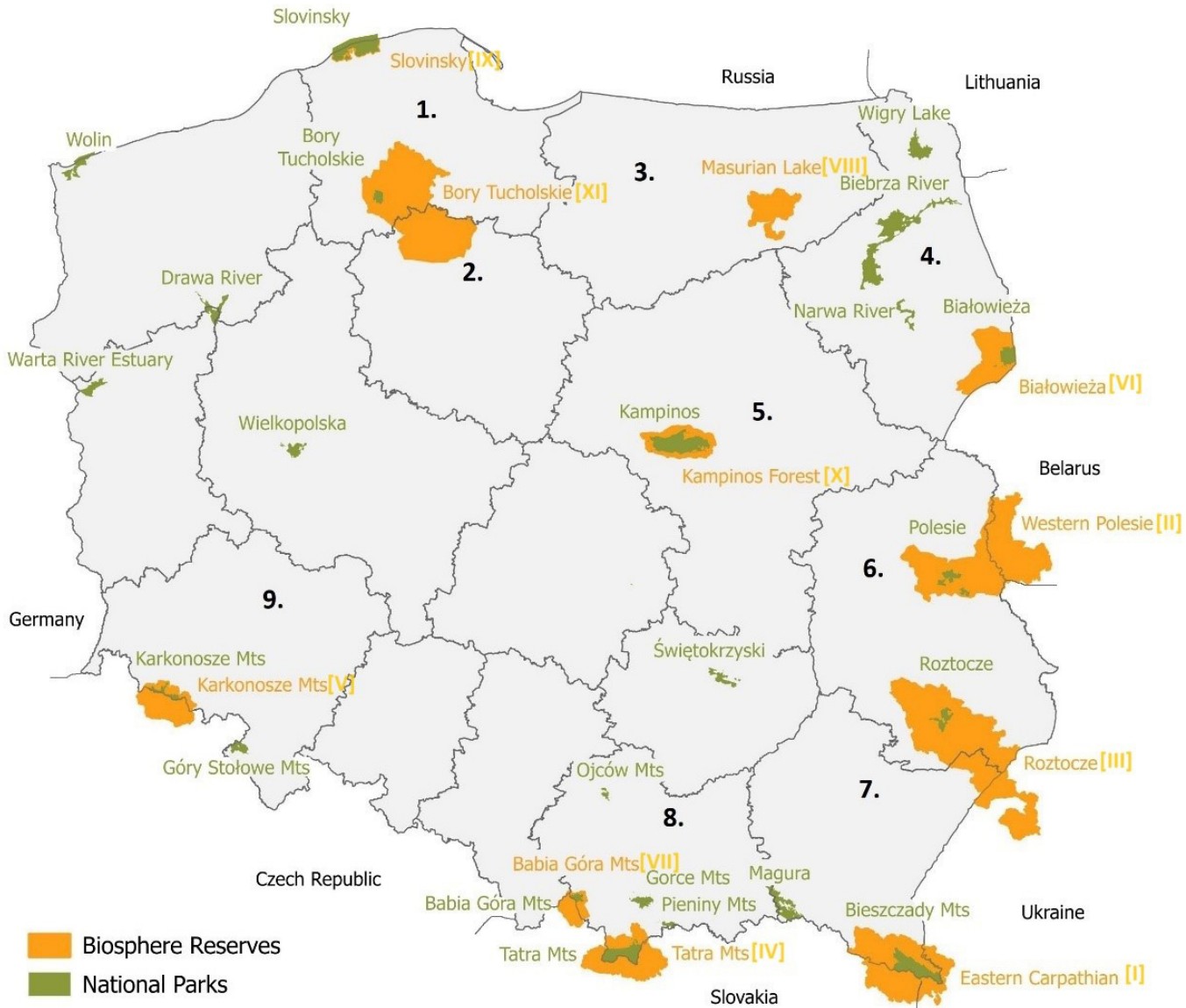

**Figure 1.** Location of biosphere reserves in Poland (including transboundary BRs); voivodeships: 1. Pomeranian, 2. Kuyavian-Pomeranian, 3. Warmian-Masurian, 4. Podlaskie, 5. Masovian, 6. Lublin, 7. Subcarpathian, 8. Lesser Poland, 9. Lower Silesian; [I]–[XI]—the numbers in the figure correspond to the BRs in Table 1. (by authors).

**Table 1.** General information about Biosphere Reserves (BRs) in Poland (by authors).

| No | Biosphere Reserve | National Forms of Nature Conservation | Year of Appointment | Surface | | |
| | | | | Total (Core, Buffer, Transition Zone) | Core Zone | |
| | | | | ha | ha | % |
| Transboundary Biosphere Reserves | | | | | | |
| I. | Eastern Carpathian Mts International BR (Poland-Slovakia-Ukraine) | Poland–Bieszczady National Park, Natural Landscape Parks: Ciśnieńsko-Wetliński, San Valley; Ukraine–Użanski National Park, Nadsansky Regional Landscape Park; Slovakia–Poloniny National Park | 1992/1998 | 109,694 | 18,536 | 16.9 |
| II. | Western Polesie Transboundary BR (Poland-Belarus-Ukraine) [1] | Poland–West Polesie Biosphere Reserve: Polesie National Park with a buffer zone, Sobiborski Landscape Park; Ukraine–BR of Shatsk Raion: Szacki National Park; Belarus–"Polesie Nadbużańskie" BR | 2012 | 139,917 (Poland) | about 5200 (Poland) | about 3.7 (Poland) |
| III. | Roztocze Transboundary BR (Poland-Ukraine) | Poland–core zone: Roztocze National Park, Nature Reserves: Św. Roch. Debry, Bukowy Las, Nad Tanwią, Czartowe Pole, Jalinka and Źródła Tanwi; buffer zone: Landscepe Parks: Szczebrzeszyn LP, Krasnobród LP, South Roztocze LP, Solska Forest (with nature reserves: Zarośle, Sołokija); part of the buffer zone of Roztocze National Park; Ukraine–core zone: Roztocze Nature Reserve, Yavoryvskyi National Nature Park, Nemyriv Nature Reserve, some areas of Roztocze Rawskie Regional Landscape Park; buffer zone: buffer zone of Roztocze Nature Reserve, recreation area of Yavoryvskyi National Nature Park, Roztocze Rawskie Regional Landscape Park | 2019 | 371,902 297,015 (Poland) 74,887 (Ukraine) | 12,474 9149 (Poland) 3325 (Ukraine) | 3.4 3.1 (Poland) 4.4 (Ukraine) |

**Table 1.** *Cont.*

| No | Biosphere Reserve | National Forms of Nature Conservation | Year of Appointment | Surface | | |
|---|---|---|---|---|---|---|
| | | | | Total (Core, Buffer, Transition Zone) | Core Zone | |
| | | | | ha | ha | % |
| IV. | Transboundary BR of Tatra Mts (Poland-Slovakia) | Poland–Tatra Mts National Park; Slovakia–Tatra Mts National Park with buffer zone | 1992 | 126,056 20,396 (Poland) 105,660 (Slovakia) | 56,992 7548 (Poland) 49,444 (Slovakia) | 45.2 37 (Poland) 46.8 (Slovakia) |
| V. | Karkonosze Mts BR (Poland-Bohemia) | Poland–Karkonosze Mts National Park with buffer zone; Bohemia–Karkonosze Mts National Park with buffer zone | 1992 | 71,799 16,830 (Poland) 54,969 (Bohemia) | 9636 1717 (Poland) 7919 (Bohemia) | 13.4 10.2 (Poland) 14.4 (Bohemia) |
| | | | Polish biosphere reservates | | | |
| VI. | Białowieża Forest BR | Białowieża Forest National Park | 1976, extension in 2005 | 92,399 | 21,946 | 23.8 |
| VII. | Babia Góra BR | Babia Góra National Park | 1976, loss of status in 1997, regained status in 2001 | 11,829 | 1062 | 9 |
| VIII. | Masurian Lakes [2] BR | Łuknajno Lake Nature Reserve, Masurian Lakes Landscape Park | 2017 [2] | 57,600 | | |
| IX. | Slowinski BR | Slowinski Biosphere Reserve | 1976 | | | |
| X. | Kampinos Forest BR | Kampinos Forest National Park with buffer zone | 2000 | 76,232 | 5675 | 7.4 |
| XI. | Bory Tucholskie BR | Bory Tucholskie National Park, Landscape Parks: Tucholski, Wdecki, Wdzydzki, Zaborski | 2010 | 319,500 | | |

[1] In 2002, the West Polesie Biosphere Reserve was established in Poland, which was incorporated into the West Polesie Transboundary Biosphere Reserve in 2012. [2] In 1976, the Łuknajno Lake Biosphere Reserve was established in Poland with a total area of 1410 hectares, including a central core zone of 710 hectares and other parts totaling 700 hectares. The Łuknajno Lake Biosphere Reserve became part of the Masurian Lakes Biosphere reserve, which encompasses the areas of the Masurian Lakes Landscape Park, including 11 nature reserves for which buffer zones have been designated.

To this end, a review of document resources and legal, administrative, and development plans and programs was conducted to determine the occurrence and extent of information about biosphere reserves (formal, managerial, and executive). The review included:

- Legal regulations (conventions, international agreements, laws, and regulations)
- Strategic documents related to state policy on environmental protection and natural resource management as well as regional policy
- Planning documents at the national, regional, and local levels
- Nature protection plans for national parks, landscape parks, and nature reserves
- Protective plans for NATURA 2000 areas
- Economic-protective programs for forest promotional complexes

- Forest management plans for forest districts
- Audit reports conducted by the Supreme Audit Office

The review did not include documents submitted by Poland to UNESCO in connection with applications for recognition of biosphere reserves; this documentation is not publicly available.

In the review article, we analyze planning and strategic documents that should constitute the existence of biosphere reserves and enable the achievement of their primary goal, which is the coexistence of humans and the environment without excessive civilization pressure and limitations on socio-economic development. The overall objective is to demonstrate whether Poland, through BR, is achieving the goals of the Man and Biosphere program. Through this approach, sustainable development goals can be effectively implemented, including the restoration of ecosystem functionality and the satisfaction of social and economic needs in ecologically valuable areas, while ensuring the sustainability of natural systems.

## 3. Results

### 3.1. The Results of Analysis of National Legal Acts—BRs as a Tool for Legal Protection of Nature in Poland

The first Polish biosphere reserves (the Białowieża Forest Biosphere Reserve, the Babia Góra Biosphere Reserve, and the Słowiński Biosphere Reserve) were established in 1976 during the period of the first post-war nature protection act [6], which was repealed in 1991. It did not, of course, have any references to biosphere reserves, nor even the possibility of creating transboundary forms of nature protection. It only stipulated that the Minister of Forestry, as the supreme authority for nature protection, could commission the State Council for Nature Conservation to cooperate with nature conservation organizations in international relations, in consultation with the Minister of Foreign Affairs.

The next nature protection act in force from 1991 to 2004 [7] clearly stated that a national park, nature reserve, landscape park, and non-living nature documentation site can obtain international status as defined by appropriate international conventions or resolutions of international organizations, which especially concern biosphere reserves. However, their legal definition was not provided in the statutory law. Border areas of natural value could also be designated in cooperation with neighboring countries for joint protection. However, as a result of the amendment of this nature protection act in 2000, new legal provisions were presented, in which the reference to biosphere reserves was removed. In the currently applicable nature protection act [8], the legislator did not transpose biosphere reserves as a form of nature protection into Polish law, as was done, for example, in the case of NATURA 2000 areas, introduced by European Union directives [9,10]. Therefore, biosphere reserves in Poland do not have any legal basis not only in the nature protection act but also in any other legal act.

In conclusion, the analysis of legal acts has shown that the term "biosphere reserve" is not a legal category in Poland. Granting biosphere reserve status is not related to the introduction of new protective regulations in a given area but is only a kind of international recognition and promotion by UNESCO. In this sense, the term "reserve" can be very misleading in Polish conditions, as one of the legal forms of nature protection in Poland is a "nature reserve" [8]. In the social mentality and among many politicians and government and local government officials, the stereotype of a "reserve" as an area that is excluded from economic activity (e.g., forestry, agriculture) and inaccessible to the community is entrenched. It is an area that becomes "useless" and leads to the economic devaluation of the local population. Meanwhile, the biosphere reserve, in its assumption, is supposed to be a form of recognition of the efforts of local communities and the scientific community to maintain harmony between high natural values and their sustainable use. The functioning of biosphere reserves is not legally established, not only in Polish law but also internationally. The protection of an area recognized as a biosphere reserve does not result from a legal international convention but from an international research program based

on the non-binding agreements of UNESCO bodies [1]. It was assumed that the forms of nature protection included in the biosphere reserve would have high prestige. Thus, their protection would be more careful than other types of nature protection. Meanwhile, in Poland, there is no formal and legal distinction between national parks, nature reserves, and landscape parks that are part of biosphere reserves and other national parks, nature reserves, and landscape parks.

*3.2. Results of National (Polish) Strategic Analyses—The Impact of BRs on Local Development and Social Initiatives*

3.2.1. National Strategic and Planning Documents

At the national level, there is a lack of information on biosphere reserves in strategic documents such as the "National Strategy for Regional Development 2030. Socially Sensitive and Territorially Sustainable Development" [11], the "Ecological Policy of the State 2030—development strategy in the field of the environment and water economy" [12], and the "Strategy for Responsible Development up to 2020 (with a perspective until 2030)" [13]. Similarly, there is no information on biosphere reserves in the "Program for the Protection of Sustainable Use of Biological Diversity along with the Plan and Actions for the years 2015–2020" [14]. Only when discussing the problem of increasing Poland's participation in international forums for biodiversity protection was attention drawn to the protection and management of species in transboundary regions. The "Concept of Spatial Development of the Country 2030" [CSDC 2030] [15] refers, to the greatest extent, an idea or principles of BRs. When presenting the vision of spatial development of Poland, it was emphasized that the Polish space, preserving the richness of natural and cultural heritage values, is clearly recognizable. According to CSDC 2030, the basis for the protection of the most valuable natural and landscape resources was to be, among others, the integration of activities in the field of designation, functioning, and protection of a coherent system of protected areas, including through the progressive integration of the management of areas belonging to various networks, including the planned expansion of the biosphere reserve network in the country, which already includes some national parks [15]. These activities were to include the establishment of transboundary protected areas with different functional and spatial programs and different levels of protection. It was assumed that it would be necessary to establish the Suwałki-Wisztyniecki Park (Poland, Lithuania, Russia), the Vistula Spit and Vistula Lagoon Park (Poland, Russia), and the bilateral (Poland, Slovakia) Pilsko Mountain Reserve [Żywiec Beskids, Orava Beskids (Oravské Beskydy)]. It was also planned to expand the national network of biosphere reserves by creating the Tri-National Three Forests Biosphere Reserve (Poland, Lithuania, Belarus), the Lower Odra Valley Biosphere Reserve, the International Białowieża Forest Biosphere Reserve, the Bug Valley corridor with the Transboundary Bug Gorge Protected Area (Poland, Belarus), the Tri-National Western Polesie Biosphere Reserve (Poland, Belarus, Ukraine), the International Roztocze Biosphere Reserve (Poland-Ukraine), and the enlargement of the Babia Góra Biosphere Reserve. Only the Białowieża Forest Biosphere Reserve, the Roztocze Transgenic Biosphere Reserve, and the Western Polesie Transgenic Biosphere Reserve were established by 2019. Unfortunately, this document (CSDC 2030) was repealed in 2020 [16].

Additionally, our proprietary analysis of the resources of the Central Register of Nature Protection Forms, maintained by the government agency, namely the General Directorate for Environmental Protection, has shown that information on the establishment of international protection in the form of a biosphere reserve is provided for all 10 national parks covered by it. However, there is a lack of such information for 12 landscape parks, which are also covered by the MaB program.

3.2.2. Regional and Local Strategic and Planning Documents

At the regional (voivodeship) level (NUTS 2), very fragmentary information on biosphere reserves is included only in the development strategies of three voivodeships (Masovian, Subcarpathian, and Lesser Poland voivodeships) [17–19] out of nine that have

BRs [19–25]. However, there are provisions regarding BRs in the spatial development plans of all voivodeships. These provisions are a derivative of information on biosphere reserves contained in the no longer applicable CSDC 2030 [15]. They are included in the following paragraphs of the spatial development plans of the voivodeships:

- internal development conditions of spatial development–voivodeships: Podlasie (Białowieża Forest Biosphere Reserve), Warmian-Masurian (Masurian Lakes Biosphere Reserve), Pomeranian (Słowiński Biosphere Reserve, Bory Tucholskie Biosphere Reserve) [26–28],
- state and directions of spatial development: Masovian voivodeship (Kampinos Forest Biosphere Reserve) [29], Biosphere Reserve Babia Góra, Transgenic Biosphere Reserve Tatra Mts. [30],
- directions of development and spatial policy: Subcarpathian and Lublin (Roztocze Transgenic Biosphere Reserve, West Polesie Transgenic Biosphere Reserve) [31,32],
- methods of rational and sustainable use of natural, cultural, and landscape resources: Lower Silesian voivodeship (Biosphere Reserve of Karkonosze Mts) [33],
- functional areas of supra-regional importance: Kuyavian-Pomeranian voivodeship (Bory Tucholskie Biosphere Reserve) [34].

Therefore, there is a lack of a coherent system for treating and informing about the presence and role of BRs in planning documents at the voivodeship/region level. The Lesser Poland Voivodeship Spatial Development Plan emphasizes the priority of spatial development in the nodal areas that are part of the biosphere reserves: the Babia Góra Biosphere Reserve and The Transboundary Biosphere Reserve of Tatra Mts.

At the local level (municipalities and local administrative units (LAUs)), there is a lack of information about biosphere reserves in municipal planning documents (land use studies and plans). This may be due to the fact that local planning documents are closely related to regional documents. In the absence of references to biosphere reserves in regional documentation, it is not surprising that there are no mentions at the level of local documents.

There is currently no coherent spatial management in Poland. This is mainly due to the lack of a document specifying the spatial organization of the country and the lack of references and connections between the national and regional levels. Incomplete connections between spatial planning documents only occur at the region and commune level. The study of conditions and directions of the spatial development of the commune (currently the general plan) defines, among others, establishing a spatial development plan for the region (voivodship). Plans at the commune level cannot be inconsistent with the arrangements contained in the general plan, but their implementation is not obligatory.

### 3.2.3. Plans for Protecting Nature Forms Including BRs

There is also a lack of detailed information on biosphere reserves in the protection plans of national parks, i.e., documents that legally define protection goals, protection tasks, and requirements for local spatial planning documents [8,35]. If they appear at all: the Bieszczady National Park [36], the Kampinos National Park [37,38], the Białowieża National Park [39], Polesie National Park [40], Karkonosze National Park [41]; they usually provide general information, e.g., the most important and touristic popular Polish mountains park, Tatra National Park. "Both national parks, Polish and Slovak, constitute the Tatra Transboundary Biosphere Reserve" (Chapter 1, point 3.4) [42]. This is the only sentence in the whole Tatra NP protection plan referring to the biosphere reserve. The protection plans for the Bory Tucholskie National Park [43] and the Roztocze National Park [44] were adopted before the biosphere reserves were established (the Bory Tucholskie Biosphere Reserve, the Transboundary Roztocze Biosphere Reserve), and have not yet been updated with respect to the presence of biosphere reserves.

The Babia Góra Biosphere Reserve receives relatively more attention in the protection plan for the Babia Góra National Park [45]. In discussing the social and natural conditions for achieving conservation goals and the social and natural conditions for implementing

them, attention was paid, among other things, to the fact that the proper functioning of the biosphere reserve depends on cooperation between the park, local governments, associations, and units operating in the municipalities. This case of good cooperation between the national park management and local authorities, especially for the development of tourism and the promotion of the region, can be considered exemplary.

The biosphere reserve was also taken into account in the requirements for studies of conditions and directions for spatial development of municipalities, local spatial development plans, and spatial development plans for the Lesser Poland and Silesian voivodeships, concerning the elimination or limitation of internal and external threats. The requirements for the spatial development plan for the Lesser Poland voivodeship and the requirements for the spatial development plan for the Silesian voivodeship proposed, among other things, to include the ranges and goals of nature protection forms and areas related to nature protection, also in the case of biosphere reserves. Similarly, the requirements for the study of conditions and directions for spatial development and the local spatial development plan in the municipalities of Zawoja, Lipnica Wielka, and Jabłonka proposed to include the ranges and goals of nature protection forms and areas related to nature protection, not only in the Babia Góra National Park and its buffer zone, NATURA 2000 areas (Babia Góra, Ostoja Babiogórska), and the neighboring Slovakian protected natural areas but also the Babia Góra Biosphere Reserve.

There is a lack of information about biosphere reserves in plans for the protection of nature reserves and landscape parks located within their area. The only information provided is in the Protection Plan for the Masurian Landscape Park [46], which outlines the conditions and directions for spatial development studies in municipalities. The plan includes information about the existence of the Łuknajno Lake Biosphere Reserve (currently within the boundaries of the Masurian Lakes Biosphere Reserve) in a designated natural and landscape zone, and various measures for the zone. For Łuknajno Lake, these measures include scientific, educational, natural, and landscape, and tourist functions (accessibility through nature trails, pedestrian and cycling paths, and observation towers), conversion of the Łuknajno manor into an Environmental Education Centre, a ban on commercial fishing, and agricultural management based on agro-environmental programs.

### 3.2.4. Economic Documents of Forest Area Managers Covered by BRs

Additionally, fragmented information about biosphere reserves is included in several Economic and Protective Programs for Forest Complexes (LKP: Tuchola Forest, Bieszczady Forest, Masurian Forests, Białowieża Forest). The Bieszczady Forest LKP is entirely located within the Eastern Carpathian Biosphere Reserve. Forest Promotional Complexes in Poland are designed to promote sustainable forest management, conduct scientific research, and provide forest education to the public.

### 3.3. Results of the Analysis of Documents Evaluating the Functioning of Biosphere Reserves

So far, the Polish supervisory body, the Supreme Audit Office (NIK), has taken control actions only in the case of one biosphere reserve. These actions covered the functioning of the Eastern Carpathian Biosphere Reserve (ECBR) in the years 2012–2016 and were part of an international audit carried out by audit institutions in Slovakia [Supreme Audit Office (NKU)] and Ukraine [47]. This audit, carried out on the basis of agreements concluded between individual audit bodies, was carried out as part of the European Organisation of Supreme Audit Institutions (EUROSAI). After the audit was completed, NIK did not formulate any post-audit conclusions directly related to the functioning of the ECBR. This was not possible because the biosphere reserve has not yet been introduced into the national legal order and has no legal definition.

On the Polish territory of the Eastern Carpathian Biosphere Reserve, the functions of the biosphere reserve were carried out by the national park and landscape parks. The audit showed that the Bieszczady National Park, which constitutes the central zone of the Eastern Carpathian Biosphere Reserve, carried out its statutory tasks not on the basis of the

existing protection plan, but based on protective tasks established every year. Although none of them were defined as tasks of the biosphere reserve, the Supreme Audit Office (NIK) assessed that the activities of the Bieszczady National Park were, in fact, identical to the tasks and recommendations for the biosphere reserve in the so-called 'Pamplona Recommendations'.

Although the tasks related to nature protection were carried out in the strict protection zone of the Bieszczady National Park, in accordance with national regulations governing national parks, they did not result from any common strategies or coordinated actions aimed at protecting the entire central zone of the Eastern Carpathians Biosphere Reserve. According to the NIK's assessment, the lack of such strategies means that the "Recommendations from Pamplona" regarding the development of common or coordinated protection strategies, which should be implemented in the central zone of the biosphere reserve (e.g., protection strategies for migrating animal species), have not been fully implemented, nor are they being implemented to this day. Similarly, on the Slovak and Ukrainian sides, individual zones of the Eastern Carpathian Biosphere Reserve operate solely on the basis of national legislation concerning nature protection forms.

The statutory tasks of the landscape parks were included in annual work plans, approved by the Board of the Subcarpathian Voivodeship, and in the protection plan for the Ciśniańsko-Wetliński Landscape Park, as well as in the resolution of the Subcarpathian Voivodeship Council, which defined, among other things, special objectives for natural and cultural heritage protection of this park in the absence of a protection plan for the San Valley Landscape Park. These documents did not provide for the implementation of biosphere reserve tasks. However, these activities also corresponded to the catalogue of tasks serving the implementation of the goals and functions of the biosphere reserve, recommended in the "Recommendations from Pamplona" [47].

According to the Supreme Audit Office (NIK) report, the cooperation between all Polish parks within the biosphere reserve and the cross-border cooperation with parks on the Slovak and Ukrainian sides had a more formal than practical dimension. These parks did not collaborate in planning and carrying out statutory tasks and did not undertake any joint initiatives that would be implemented only for the International Biosphere Reserve of Eastern Carpathian Mts. There was also no joint execution of tasks in a cross-border dimension, despite the declaration of joint action by each of the parties forming the biosphere reserve. Currently, under the conditions of the war in Ukraine, there are no opportunities for cooperation, due to the focus on other priorities.

In terms of implementing the goals and functions of the biosphere reserve, all of the controlled parks actively cooperated with local government authorities. This cooperation mainly concerned issues related to the spatial planning of municipalities. In the NIK's opinion, both the cooperation of the Bieszczady National Park (as the central and buffer zones) of the biosphere reserve, as well as the landscape parks (as the transition zone), with these entities fully ensures the implementation of all tasks specified in the Framework Statute of the World Network of Biosphere Reserves.

The lack of legal grounds for the existence of a biosphere reserve in national law prevents the establishment of a management body with the competencies to coordinate or manage all activities carried out in the biosphere reserve area. In the case of the Transboundary Biosphere Reserve of East Carpathians, the coordinating council established by the directors of the parks that make up this biosphere reserve is only an informal body without any legal basis in the legislation of Poland, Ukraine, and Slovakia. It is purely declarative and has no specific competencies and powers allowing for effective management and coordination of activities both in the entire biosphere reserve area and in its individual parts (Polish, Ukrainian, Slovak). However, it allows the managing directors of protected areas to exchange information, consult documents, and undertake various initiatives (e.g., joint publications, educational programs, cultural events, and nature trails). In the current situation, with the existing border protection system of the Schengen area separating the Ukrainian part from the Polish and Slovak parts, different regulations in national biosphere

reserve areas and resulting competencies, and different financing systems, the established coordinating council is unable to meet the requirements set for a coordinating body in the Pamplona Recommendations. Its managerial capabilities are practically limited. On the other hand, cross-border BRs in Slovakia are involved in the development of international cooperation within the Global BRs Network [48].

Regardless of this, according to NIK, a parallel and necessary action should be a legislative initiative aimed at introducing regulations concerning the biosphere reserve into the national legal order, including providing a legal definition of this concept. According to NIK, this would enable the unambiguous determination of the financial means necessary to carry out tasks in individual countries by all units comprising the biosphere reserve. It would also enable the pursuit of funding from other sources, including the European Union, and its acquisition would enable the real implementation of joint tasks by partners from Ukraine, Slovakia, and Poland.

The results of the NIK audit are consistent with the findings and recommendations of the Slovak and Ukrainian control bodies, which audited the parks co-creating the Eastern Carpathian Mts Transboundary Biosphere Reserve in their countries. These bodies directed their recommendations to the Government of Slovakia and to the Supreme Council of Ukraine. There is no information on the practical results of these audits. The results of the audit conducted by NIK on the Eastern Carpathian Mts International Biosphere Reserve were also presented at a meeting of the Environment Committee of the Senate of the Republic of Poland. The discussion held during this meeting, repeatedly emphasized the lack of the term "biosphere reserve" in Polish law and the resulting consequences, but to date, no legislative initiatives or changes in the law related to biosphere reserves have arisen.

None of the government agencies authorized to do so (Ministry of Climate and Environment, previously Ministry of Environment, General Directorate for Environmental Protection) have provided detailed information on biosphere reserves. The report on the state of biodiversity in Poland during the 10 years after the first Earth Summit in Rio de Janeiro in 1992 only provided brief information on transboundary areas, including biosphere reserves, the creation of which is related to the implementation of provisions arising not only from international conventions but also from other agreements [49].

There are no references to biosphere reserves in the Carpathian Convention established on the basis of treaty law [50], which defines a comprehensive policy for the protection and development of the Carpathians, including the preservation of natural conditions and cultural heritage. This also applies to thematic protocols adopted under this Convention (Protocol on the protection and sustainable use of biological and landscape diversity, sustainable tourism, sustainable forest management, sustainable transport, sustainable agriculture, and rural development). Meanwhile, all Protocols oblige the Parties to the Convention (the Czech Republic, the Republic of Poland, Romania, the Republic of Serbia, the Slovak Republic, and Ukraine) to take joint action throughout the Carpathians and by individual parties on their own territory based on relevant provisions of national law.

## 4. Discussion

The significance of public and political attention lies in its role in transforming biosphere reserves into recognized and endorsed regions, referred to as "promoted areas" [51]. These areas are characterized by having their objectives incorporated into various policies and actively promoted. In contrast to national parks, which prioritize the preservation and regeneration of nature, biosphere reserves adopt a more comprehensive and inclusive approach, as highlighted by [52]. This means that Biosphere Reserves combine nature conservation with sustainable social, cultural, and economic development. However, ref. [53] suggested that the legal borders of protection areas have been fictitious, which undermines the premises of the successive environmental protection, because the pressures on the boundary of land use overlap with the legal environmental restriction and land use will never fulfil the goals established for the protected areas. Despite occasional conflicts within

their areas, real actions are being taken to combine the protection of biological and cultural diversity with economic and social development in accordance with the principles of sustainable development (e.g., [54–63]). However, the situation in Poland is clearly different, as indicated by a number of documents concerning spatial planning and management, regional and local development, and nature conservation.

The Biosphere Reserve, in its essence, is meant to recognize the efforts of local communities and the scientific community in maintaining harmony between high natural values and sustainable use. The opinion of local communities regarding actions undertaken by administrative bodies and authorities is particularly important. Social acceptance of conservation actions plays a fundamental role in achieving social consensus, implementing sustainable development, and ultimately succeeding in conservation efforts [56,57,63]. The analysis of materials may indicate a lack of government willingness to promote the idea of biosphere reserves. This is supported by the analysis, which points to the lack of legal provisions for the concept of BRs in Polish legislation and the absence of references to BRs in the government's adopted Strategy for Sustainable Development. Analyses have also shown that experts involved in strategic and planning activities or influencing the enactment of laws (legislation) do not take into account biosphere reserves in their prepared documents and are not aware of the role of BRs in preserving not only natural, but also cultural, social, and economic heritage. The active engagement of society, entrepreneurs, and scientific experts is crucial as they should be involved as stakeholders and advocates of the MaB concept and its implementation, particularly in the management of biosphere reserve (BR) areas. The overall awareness and understanding of BRs among various social groups are currently lacking, as evidenced by the absence of BRs in local planning documents. This knowledge gap extends to experts involved in preparing planning documentation, local authorities, and local communities.

To effectively support long-term biodiversity management, it is important to provide support to landowners in their management efforts and foster collaboration among diverse stakeholder groups, including those who own protected lands. This collaborative approach, involving networking and cooperation, can prove more effective in achieving sustainable biodiversity outcomes. In addressing the dynamic nature of environmental challenges, it is essential to adopt diverse approaches that encompass multiple instruments beyond just legal and planning mechanisms. This adaptive policy response, as emphasized by [64], allows for flexibility and innovation in tackling complex environmental issues.

Although the most information about biosphere reserves was included in "Concept of Spatial Development of the Country 2030", this document was repealed in 2020 [16].

Voivodeship spatial development plans have marginal significance because they do not constitute a coordinating instrument for other planning documents at the national, functional area, or local levels. The system of mutual agreements between voivodeship plans and local plans does not work at all because creating local plans is not mandatory.

Since under Polish law biosphere reserves can only be recognized as an informal form of nature protection organization, rather than a legally regulated form of nature protection, legal terms related to the functioning of biosphere reserves cannot appear in documents such as national park protection plans, landscape park protection plans, or nature reserve protection plans. They also cannot be included in protection tasks for national parks and nature reserves until a protection plan is established. Action items defined nominally as "biosphere reserve tasks" cannot be included in protection plan projects or protection task projects, because the content of these documents is strictly defined by the Nature Protection Act [8] and the relevant executive decree [35]. There are no formal commitments and promises regarding the sources of financial, material, and human resources for implementing biosphere reserve goals, as exemplified by the Roztocze Biosphere Reserve project [65]. The assumption in the project that every institution located in the biosphere reserve area that is to be involved in its management and development (local governments, forest districts, regional environmental protection directorates, NGOs, national parks) will include in their annual budget financial resources for the implementation of biosphere

reserve goals and related projects is incorrect. Financial resources cannot be included in a budget for something that does not formally exist.

The assumption that the functioning of a biosphere reserve results from the goodwill of all entities managing and administering it, taking into account equal partnership between them [65], is naive in the Polish reality. Different legal regulations and rules governing the use and access to the different zones of BRs (core, buffer, transition) apply. The ownership of biosphere reserve lands is diverse (although state-owned lands predominate), as is the ownership structure: private, under the supervision of State Forests or local governments (county governments), municipal, church, and religious associations. Users of the zones are not only very diverse (state and local administration, local communities, State Forests, landscape parks (landscape park administrations), regional environmental protection directors), but they also have different competencies. Their tasks are often divergent.

The implementation of the MaB program's objectives becomes challenging due to the involvement of numerous entities in managing the biosphere reserve. Previous research conducted through case studies in the Czech Republic, Hungary, and Poland revealed several key obstacles. Firstly, there was a lack of vertical integration between MAB institutions and national authorities. Additionally, there was a shortage of adequately trained personnel, insufficient funding, and limited political support for local implementation [66].

However, the example of a peripheral biosphere reserve in Bavaria, Germany, demonstrates that local politicians place less emphasis on the biosphere reserve's contribution to economic development [67]. Research on policy diffusion through governmental channels has shown two typical outcomes: the policy is either adopted in a copy-and-paste manner or adapted to suit local conditions [68].

In the case of Japanese and Russian biosphere reserves, three main factors influencing the level of local involvement have been identified: the historical relationship between the government and local communities, perceptions of nature protection, and attitudes towards the economic benefits derived from nature, that is geo- and eco-tourism and tourism-related services, thanks to unique natural sites with geological objects, cultural landscapes, and historical sites [69].

Therefore, our assertion, supported by the existing literature (e.g., [70–73]), is that apart from legislative solutions, it is imperative to educate various stakeholders, including societies, local authorities, and employees of institutions responsible for managing biosphere reserve areas. This is necessary because the natural and cultural values of protected landscapes are often neglected in the practical work of biosphere reserves [74].

The aim of the current provisions of the Carpathian Convention [50], which apply to an area of approximately 6% of the Polish's land surface in the Lesser Poland, the Subcarpathian, and the Silesian voivodeships (18,612.48 km$^2$), is international cooperation to improve the quality of life, strengthen the local economy and communities, and preserve the natural, landscape, and cultural heritage values of the Carpathians.

The Carpathian Network of Protected Areas (which presumably includes all spatial forms of nature protection existing and created in Poland, Slovakia, and Ukraine in the Carpathians) is not fully developed in Poland, and valuable Carpathian areas are irreversibly destroyed. This is particularly true for areas in the upper course of the Wiar (Wihor) river in the Przemyśl Foothills (Sanok-Turka Mountains), where for almost 40 years, the establishment of the Turnicki National Park has been planned to protect the surviving fragment of the former Carpathian Forest [75–77]. Irreversible damage and destruction are being caused. State Forests and local residents have been opposing its creation for years. Not only are there no actions being taken to formally establish this national park, but the intensity of threats to natural resources, including widespread deforestation and the destruction of aquatic and riparian habitats through poorly conducted technical investments, is increasing year by year. In the current situation (war), cross-border cooperation with Ukraine is currently limited. As a result of the war, natural resources, especially in the southern and eastern parts of Ukraine, are being destroyed [78].

Other countries have similar experiences. In Korea, some areas face challenges in obtaining biosphere reserve designation due to opposition from local residents [79]. To overcome such obstacles, managers need to prioritize community participation, which plays a crucial role in fostering compliance with regulations that support protected areas. This can be achieved by improving communication among stakeholders, encouraging active participation, and promoting capacity development [66,80,81].

Understanding pro-environmental behaviors within protected areas is essential for mitigating negative compounded impacts and maximizing positive impacts [82]. Among various forms of participation, a practice-based approach appears to enhance legitimacy of protection. Research conducted on 92 Biosphere Reserves (BRs) across 36 countries through a two-wave survey in 2008 and 2013 [83] suggests that the relationship between participation and the legitimacy of nature reserves in local communities is not linear. Moreover, increased levels of participation do not necessarily lead to a higher level of legitimacy for the nature reserve within the local community. Proper and full consideration of public values is a must [84], but [85] shows that there are major deficiencies in expert-based designation processes in terms of their ability to reflect the views of the wider public about what they consider to be important and in need of protection.

However, protective and managerial tasks are carried out jointly, without dividing them into the BRs zone and national or landscape parks. Controls conducted in the Polish, Slovak, and Ukrainian parts of the Eastern Carpathian Mts International Biosphere Reserve clearly showed, however, that the main difficulty in international cooperation within this biosphere reserve is the significantly different legal systems governing the operation of protected objects (nature protection forms) and different land ownership relationships in these three countries. As a result, different ways of managing natural resources have emerged. A clearly positive solution is the development of a common protective zoning strategy and the exchange of scientific and practical experiences in nature protection throughout the International Eastern Carpathian Mts Biosphere Reserve area. The rank and prestige on the international stage are also important, although this does not necessarily translate into national policies for protecting natural resources. In the Polish section of the International Eastern Carpathian Mts Biosphere Reserve, there has been a lack of cooperation between national and landscape parks in planning and implementing the statutory tasks within the biosphere reserve area [86–91]. Additionally, there has been a lack of joint projects between the parks that co-create the biosphere reserve on the Slovak and Ukrainian sides [48]. However, the analysis of activities from the southern border indicates the dominance of the main services for the tourism industry, as well as the relatively high participation in advanced scientific programs conducted by universities and research institutions [92].

The absence of legal obligations to cooperate arises from the fact that the tasks of the biosphere reserve are not defined in national laws or any documents governing the operation of these parks in respective countries. This absence of a legal framework and legislative differences between Poland and Slovakia [93], as well as differences in the interpretation of various forms of protected areas (e.g., national parks, landscape parks, and nature reserves), can complicate the coordinated management of transboundary biosphere reserves.

## 5. Conclusions

1.  Polish biosphere reserves are not established in Polish legislation and contrary to UNESCO recommendations, cannot be subject to it. For over 45 years, since the first BR was adopted in Poland, there have not been legal forms of nature protection. They have not contributed to increased conservation and sustainable development efforts in the country, whatever that might mean. The idea of sustainable functioning of nature and humans is generally either little known or completely alien.
2.  The significance of BRs in Polish political and socio-economic realities is only declarative. Generally, biosphere reserves do not have a translation into strategically planned socio-economic development, expressed by development documents (development

strategies, planning, and spatial development documents). Valuable natural areas, including forms of nature protection and protected areas based on ratified international conventions and other international agreements (e.g., UNESCO), despite the role and function they should play in nature and landscape protection, lose out to investment and settlement pressure that guarantees profits.

3. Biosphere reserves in Poland do not function according to the assumptions of the Man and the Biosphere Program and are not model examples of protected areas implementing sustainable development principles. The assessment carried out shows that the real functioning of biosphere reserves in Poland is very doubtful.

4. Revision of national legislation is absolutely necessary to provide a clear legal framework for biosphere reserves as forms of nature protection with defined tasks. Polish biosphere reserves are an untapped research field for sustainable development, ecosystem services, and experiences related to social and economic relationships between humans and the environment. It is necessary, in cooperation with scientific, social, and local government communities involved in local spatial policies, to develop an action program for nature reserves in Poland. This should be a strategic program, until 2040, and shorter operational programs, covering sequences of 5 years each.

5. The Polish side does not have the legal and financial instruments to develop transboundary cooperation on biosphere reserves. In the current situation, cross-border cooperation with Ukraine is currently very limited (war). With the existing Schengen border protection system separating the Ukrainian part from the Polish and Slovak parts, the coordinating council of the Eastern Carpathian Transboundary Biosphere Reserve is not able to meet the requirements for a coordinating body in the Pamplona Recommendations.

**Author Contributions:** Conceptualization, B.R.; methodology, B.R. and M.H.; formal analysis, B.R. and M.H.; investigation, B.R. and M.H.; resources, B.R.; data curation, B.R.; writing—original draft preparation, B.R.; writing—review and editing, B.R.; visualization, B.R.; supervision, B.R.; project administration, B.R.; funding acquisition, B.R. All authors have read and agreed to the published version of the manuscript.

**Funding:** This research received no external funding. It was made solely due to the scientific interests of the authors, who appreciate the importance of the problem of nature conservation in Poland.

**Institutional Review Board Statement:** Not applicable.

**Informed Consent Statement:** Not applicable.

**Data Availability Statement:** We used only official sources: government and regional documents and articles that are publicly available, in accordance with the provided bibliographic entries. We did not use private data; we did not create new databases. All data sources are included in "References".

**Acknowledgments:** We are thankful to the scientists and our friends of Wroclaw University of Environmental and Life Sciences for their understanding and support of our research. The views expressed here are those of the authors alone and do not necessarily reflect the common opinions in Poland.

**Conflicts of Interest:** The authors declare no conflict of interest.

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
