# Peer review of "Implementation of Biosphere Reserves in Poland–Problems of the Polish Law and Nature Legacy"

_sustainability, doi:10.3390/su152115305_

Round 1
Reviewer 1 Report
The article gives a comprehensive and very critical picture of the status and position of biosphere reserves in Poland within the framework of nature and landscape protection documentation or within spatial planning documentation at various levels, from national concepts to municipal plans. It also assesses the perception of the status of biosphere reserves by the scientific and professional community, as well as their perception by the professional public.
At the time the idea of establishing biosphere reserves was conceived, they were defined as areas where the most important parts of the biosphere would be protected and, at the same time, some of them would be used by man in accordance with the concept of sustainable use of the areas. Among other things, scientific research on these areas was also declared here, in order to obtain accurate data on the evolution of the areas and the state of the abiotic and biotic components of the natural environment and to assess the impact of human activities. This was done to pursue the possibility of guiding the further use or strict protection of biosphere reserve areas in an expedient manner.
Today, this concept has been sidelined and biosphere reserves have been relegated to the position of "registered areas" with no functional significance. They have ceased to be of any relevance in the practical work of nature and landscape conservation.
Nowadays, in addition to national nature conservation legislation, the protection of nature and its biotic components, included in the European network of protected areas Natura 2000, which is declared by the Member States of the European Union for the conservation of Europe's most valuable and endangered species and habitats, is rising in importance. However, this concept is based on different principles to those defined by UNESCO for biosphere reserves. However, the practical work of nature and landscape conservation is currently based mainly on the principles of Natura 2000, which is also declared in national legislation and in the principles of appropriate assessment of the impact of plans, programmes and projects on Natura 2000 sites.
Therefore, the assessment of the role of biosphere reserves and their position in society as a whole presented in this article is an important contribution to reviving the idea of their justification and increasing their importance in the further process of nature and landscape conservation, the protection of the most important components of the natural environment, but also in the process of using their territories for the further development of human society.
I have no conceptual or otherwise serious comments to make on the article.
Formally, it would be useful to modify Table 1 so that the texts in the first and second columns do not merge - e.g. by indenting the texts in the individual cells of the table.
Author Response
We would like to express our sincere appreciation to Reviewer 1 for reading our manuscript above. We greatly appreciate your time as reviewers and your comments, which helped us improve our article. Here is our response to your comments: Response 1.: table 1 has been corrected.
Thank you very much, Authors.
Reviewer 2 Report
The aim of the paper should be improved. The current version of it is too broad. Furthermore, it does not relate to the main research problem under consideration.
The title of the paper could be better edited.
The discussion (Chapter 4) is too extensive and descriptive. I suggest rephrasing and shortening.
In the conclusions, it was stated ‘… management of BRs and the implementation of tasks contained in the MaB program, particularly those related to sustainable development of environment-society-economy, are illusory in Poland’. The statement about illusory is exaggerated and inadequate for a scientific article. It should be written differently, reworded.
Author Response
We would like to express our sincere appreciation to Reviewer 2 for their thorough analysis of our work and all comments. We would like to stress that we have made the best effort to ensure that the corrected version of the article will meet Reviewer 2’s expectations. We tried to introduce explanations in our answers, counting on Reviewer 2’s understanding and favor. We have made changes to the text of the article that we hope will fimprove our article. Thank you for sending the reviews. Below are our mailny comments. We have highlighted all the changes made in the article in red. We have marked the parts of the text to be removed in yellow and crossed out.
| Reviewer 2 remark | Response |
|
The aim of the paper should be improved. The current version of it is too broad. Furthermore, it does not relate to the main research problem under consideration. |
We would like to thank the Reviewer for the
valuable comment. Now, the aim is: The aim of this review article is to assess to what extent international commitments and conventions regarding the Man and Biosphere program, are being realized in Poland through national legislation, economic and social development strategies, and implementing documents (spatial planning, nature conservation plans and programs). |
|
The title of the paper could be better edited. |
New title proposal: Implementation of biosphere reserves in Poland – problems of the Polish law and nature legacy. We tried to combine the suggestions of both reviewers. |
| The discussion (Chapter 4) is too extensive and descriptive. I suggest rephrasing and shortening. | We agreed that Chapter 4 is too long, so we removed the lines: 496-506; 510-515; 524-530; 580-581; 585-593. |
|
In the conclusions, it was stated ‘… management of BRs and the implementation of tasks contained in the MaB program, particularly those related to sustainable development of environment-society-economy, are illusory in Poland’. The statement about illusory is exaggerated and inadequate for a scientific article. It should be written differently, reworded. |
We would like to thank the Reviewer for the valuable comment, again. The new version of this paragraph is: management of BRs and the implementation of tasks contained in the MaB program, particularly those related to sustainable development of environment-society-economy, are illusory ineffective due to the lack of legal authorization in Poland’ |
Thank you very much, Authors.
Reviewer 3 Report
Review of
“Mismanagement of biosphere reserves in Poland – problems of the Polish law and nature legacy”
Manuscript ID: sustainability-2605758
The paper concerns legal and administrative basis for implementation/ management of biosphere (BRs) reserves in Poland.
The layout of the paper and the research design is correct. However, I have some additional comments that might help to improve the quality of the paper..
General comments:
Comment 1: Title
I suggest changing “mismanagement” to “implementation” as it seems to be more in line with the scope of the work.
Comment 2: Materials and Methods
In this chapter four research questions are raised. Taking into consideration the scope of issues discussed in the Chapter 4 (lines 617-633) I recommend to formulate an additional question concerning transboundary cooperation on relevant biosphere reserves. Afterwards, the issue should be included in the Conclusions (Chapter 5).
Comment 3: Results
In this chapter, the authors analyse a large number of studies that regulate the functioning of different types of protected areas as well as spatial planning documents adopted at national, regional and local levels. In order to facilitate readers’ understanding of the issue, I suggest to include a diagram explaining the relationship between these documents.
Comment 4: Results
Try to present the results of your analysis and evaluation of the various documents relating to the various BRs in a table.
Comment 5: Lines 193-194
Instead of "the Concept of Spatial Development of the Country 2030 (…) had the most information about biosphere reserves” I suggest you write “ CSDS refers to greatest extent to an idea or principles of BRs”
Comment 6: Lines 283-301
The Babia Gora BR is presented as an example of some cooperation between authorities of the Babia Góra National Park, regional and local governments. Can this case be considered as a model how Brs should be implemented in Poland? Try to include some evaluation.
Comment 7: Discussion, lines 435-455
This part of the manuscript should be presented in the introduction or literature review.
Comment 8: Discussion, line 496 - 500
“Voivodeship spatial development plans adopted by voivodeship assemblies have virtually no creative role. They are only a collection of content contained in other documents, conditions, and arrangements. They are a kind of inventory of what already exists in Polish legislation. Therefore, they have marginal significance because they do not constitute a coordinating instrument for other planning documents at the national, functional areas, or local level”. – It is mostly true, except for public purpose investments. Nevertheless, I consider the phrase that these development plans “have virtually no creative role” too colloquial. Try to explain it differently.
Comment 9: Discussion, line 537
“Different legal regulations and rules governing the use and access to the different zones (core, buffer, transition) apple” – do you refer to the zones designated for “nature conservation areas” ? I suggest to rephrase the sentence.
Comment 10: Discussion, line 581-582
I do not find necessary to discussed the case of the ECONET-PL as it is not the subject of the study.
Comment 11: Discussion, line 587
“Due to the dismantling of the planning system, it currently has virtually no significance”. What do you mean? Do you refer to the latest amendments to the Spatial Planning and Development Act?
Comment 12: Conclusions , line 655-656
“So far, no ecosystem service valuations have been carried out for Polish biosphere reserves” – the case of ecosystem service was not directly discussed in the study. I suggest to remove this sentence from the conclusions.
Specific comments:
Comment 1: Figure 1.
Quality of the Figure needs to be improved including colour change of the names of the BRs and National Parks. Please, add numbers of BRs on the map and Table 1, respectively.
Comment 2: Table 1
Add a column with numbers of BRs.
Comment 3: Lines 410-412
We can read that “from the discussion held during this meeting, which repeatedly emphasized the lack of the term "biosphere reserve" in Polish law and the resulting consequences, practically nothing resulted.” – In my opinion the lat part of the sentence requires proofreading.
Comment 4: line 630
You refer to publication of Mync and Szulc that was published in 1999. In this case, I suggest to refer to recent studies.
Comment 5: References
Follow the instructions for Authors concerning text editing - add reference numbers in brackets in the body text
Date of the review:
the 6th.of Otober 2023

Author Response
Reviewer 3
|
Comment 1: Title I suggest changing “mismanagement” to “implementation” as it seems to be more in line with the scope of the work. |
Thank you for your suggestion. The modification has been made. The new title: Implementation of biosphere reserves in Poland – problems of the Polish law and nature legacy |
|
Comment 2: In this chapter four research questions are raised. Taking into consideration the scope of issues discussed in the Chapter 4 (lines 617-633) I recommend to formulate an additional question concerning transboundary cooperation on relevant biosphere reserves. Afterwards, the issue should be included in the Conclusions (Chapter 5).
|
We have introduced an additional research question: 1. Is cross-border cooperation sufficient ? Are in Poland there legal and financial mechanisms to support such cooperation? And we have added a conclusion: 5. The Polish side does not have the legal and financial instruments to develop transboundary cooperation on biosphere reserves. In the current situation, cross-border cooperation with Ukraine is currently very limited (war). With the existing Schengen border protection system separating the Ukrainian part from the Polish and Slovak parts, the coordinating council of the Eastern Carpathian Transboundary Biosphere Reserve is not able to meet the requirements for a coordinating body in the Pamplona Recommendations. |
|
Comment 3: Results In this chapter, the authors analyse a large number of studies that regulate the functioning of different types of protected areas as well as spatial planning documents adopted at national, regional and local levels. In order to facilitate readers’ understanding of the issue, I suggest to include a diagram explaining the relationship between these documents. . |
We have shortened and simplified the Discussion Chapter. To clarify doubts, we added the following passage in the chapter Results: There is currently no coherent spatial management in Poland. This excludes the possibility of presenting even a simplified graphical representation of the existing exact relationships between individual planning documents. There is currently no coherent spatial management in Poland. This is mainly due to the lack of a document specifying the spatial organization of the country and the lack of reference and connections between the national and regional levels. Incomplete connections between spatial planning documents only occur at the region and commune level. The study of conditions and directions of spatial development of the commune (currently the general plan) defines, among others, establishing a spatial development plan for the region (voivodship). Plans at the commune level cannot be inconsistent with the arrangements contained in the general plan, but their implementation is not obligatory. (lines 268-278). |
|
Comment 4: Results Try to present the results of your analysis and evaluation of the various documents relating to the various BRs in a table.
|
The analysis of Polish legal acts, strategic documents regarding the country's development, planning spatial documents and international agreements (Carpathian Convention) showed that they lack information about biosphere reserves or are inaccurate. Fragmentary information about biosphere reserves appears only in spatial development plans for regions (voivodships), plans for the conservation works of Natura 2000 areas and forest management plans of forest districts. The report of the Supreme Audit Office concerns only to the International Biosphere Reserve “East Carpathian Mts". Other biosphere reserves in Poland have not been controlled so far. |
|
Comment 5: line 193–194 Instead of "the Concept of Spatial Development of the Country 2030 (…) had the most information about biosphere reserves” I suggest you write “ CSDS refers to greatest extent to an idea or principles of BRs |
We agree; the modification has been made.(line 201) |
|
Comment 6: Linie 283–301 The Babia Gora BR is presented as an example of some cooperation between authorities of the Babia Góra National Park, regional and local governments. Can this case be considered as a model how Brs should be implemented in Poland? Try to include some evaluation. |
Yes; „This case of good cooperation between the national park management and local authorities, especially for the development of tourism and the promotion of the region, can be considered exemplary”. – line 298 and the next lines |
|
Comment 7: Discussion, Linie 435–455 This part of the manuscript should be presented in the introduction or literature review. |
We agree: This part of the manuscript is presented in the introduction. |
|
Comment 8: Discussion, line 496 - 500 “Voivodeship spatial development plans adopted by voivodeship assemblies have virtually no creative role. They are only a collection of content contained in other documents, conditions, and arrangements. They are a kind of inventory of what already exists in Polish legislation. Therefore, they have marginal significance because they do not constitute a coordinating instrument for other planning documents at the national, functional areas, or local level”. – It is mostly true, except for public purpose investments. Nevertheless, I consider the phrase that these development plans “have virtually no creative role” too colloquial. Try to explain it differently. |
New phrases:“Voivodeship spatial development plans adopted by voivodeship assemblies have virtually do not inspire new planning solution. They are only a collection of content contained in other documents, conditions, and arrangements, a kind of inventory of what already exists in Polish legislation. Therefore, they have marginal significance because they do not constitute a coordinating instrument for other planning documents at the national, functional areas, or local level”. (Line 506 and the next lines) |
|
Comment 9: Discussion, line 537 “Different legal regulations and rules governing the use and access to the different zones (core, buffer, transition) apply” – do you refer to the zones designated for “nature conservation areas” ? I suggest to rephrase the sentence. |
For clarity, we have added “BR’s”:“Different legal regulations and rules governing the use and access to the different zones of BRs (core, buffer, transition) apply.” (Line 547) |
|
Comment 10: Discussion, line 581-582 I do not find necessary to discussed the case of the ECONET-PL as it is not the subject of the study. |
We agreed, so we removed the lines: 581-583. |
|
Comment 11: Discussion, line 587 “Due to the dismantling of the planning system, it currently has virtually no significance”. What do you mean? Do you refer to the latest amendments to the Spatial Planning and Development Act? |
As we have removed the above paragraph (suggestion rev. 2), this line has also been removed. |
|
Comment 12: Conclusions , line 655-656 “So far, no ecosystem service valuations have been carried out for Polish biosphere reserves” – the case of ecosystem service was not directly discussed in the study. I suggest to remove this sentence from the conclusions. |
We agree; the paragraph has been deleted |
Specific comments:
Comment 1: Figure 1. : Quality of the Figure needs to be improved including colour change of the names of the BRs and National Parks. Please, add numbers of BRs on the map and Table 1, respectively.
Comment 2: Table 1: Add a column with numbers of BRs. We improved it.
Comment 3: Lines 410-412: We can read that “from the discussion held during this meeting, which repeatedly emphasized the lack of the term "biosphere reserve" in Polish law and the resulting consequences, practically nothing resulted.” – In my opinion the lat part of the sentence requires proofreading. – added: There is no information on the practical results of these audits. The results of the audit conducted by NIK on the Eastern Carpathian Mts International Biosphere Reserve were also presented at a meeting of the Environment Committee of the Senate of the Republic of Poland. In the discussion held during this meeting, repeatedly emphasized the lack of the term "biosphere reserve" in Polish law and the resulting consequences, but to date, no legislative initiatives or changes in the law related to biosphere reserves arising. Line 427
Comment 4: line 630: You refer to publication of Mync and Szulc that was published in 1999. In this case, I suggest to refer to recent studies. We added newarticles that we managed to find, read quickly and concerned the regions we were interested in.
Comment 5: References: Follow the instructions for Authors concerning text editing - add reference numbers in brackets in the body text – done, of course, our mistake; we suggested another title of mdpi

Reviewer 4 Report
Mismanagement of Biosphere Reserves in Poland— Problems of the Polish Law and Nature Legacy is very interesting paper. Some Improvements are required.
Line 8: The study examined how the existence of BRs is reflected in Polish strategic and planning documents. (In which period is planning?)
Line 16: MaB (what is full name for MaB)/Man and Biosphere? Please to write it also in Introduction!
Line 22,23,24,25,:In conclusion, management of BRs and the implementation of tasks contained in the MaB program, particularly those related to sustainable development of environment-society-economy, are illusory in Poland. Why is not industry included in this program: sustainable development of environment-society-economy
Line 56: which aimed to develop mechanisms to support sustainable development of biosphere reserves (what are main components of biosphere reserves)
Line 85: especially transboundary cooperation (Poland-Ukraine-Slovakia?)
Line 362, 363: According to the Supreme Audit Office (NIK) report, the cooperation between all Polish parks within the biosphere reserve and the cross-border cooperation with parks on the Slovak and Ukrainian sides had a more formal than practical dimension. What is situation about the cross-border cooperation in war conditions in Ukraine? No possibility for cooperation?
Line 560: and attitudes towards the economic benefits derived from nature (Mammadova et al., 2022). What are economic benefits (an existence of some ores and critical minerals? Tourismus?)
Line 662, 663, 664: It is necessary, in cooperation with scientific, social, and local government communities involved in local spatial policies, to develop an action program for nature reserves in Poland ( in which period? For the first 5 years? Or until 2040?)
Author Response
Reviewer 4 We would like to express our sincere appreciation to Reviewer 3 for their thorough analysis of our work and all comments. We would like to stress that we have made the best effort to ensure that the corrected version of the article will meet Reviewer 3’s expectations. We tried to introduce explanations in our answers, counting on Reviewer 3’s understanding and favor. We have made changes to the text of the article that we hope will fimprove our article. To be clearly, we have highlighted all the changes made in the article in red. We have marked the parts of the text to be removed in yellow and crossed out.
| Reviewer 3 remark | Response |
|
Line 8: The study examined how the existence of BRs is reflected in Polish strategic and planning documents. (In which period is planning?) |
The study examined documents from 1947 to 2022, i.e. Polish legal acts (archived and current), the national Strategy for Responsible Development, voivodeship strategies, national park protection plans. |
|
Line 16: MaB (what is full name for MaB)/Man and Biosphere? Please to write it also in Introduction! |
The full name of the program „Man and Biosphere” is wrote on line 11. (as the first time); we added full name in the line 16.; but in the next paragraphs we used short name as „MaB” |
|
Line 22,23,24,25,:In conclusion, management of BRs and the implementation of tasks contained in the MaB program, particularly those related to sustainable development of environment-society-economy, are illusory in Poland. Why is not industry included in this program: sustainable development of environment-society-economy |
MaB areas in Poland are at the same time forms of nature protection (reserves, national parks). There is a lack of industry in these areas, and in localities located within the boundaries of biosphere reserves - intensive economic activity is very limited. The main economic activity is tourism services and sustainable forestry. Therefore, there are no references to industry in the article. |
|
Line 56: which aimed to develop mechanisms to support sustainable development of biosphere reserves (what are main components of biosphere reserves) |
The main components of RBs are a core zone and a buffer zone. We added the informations about zones in table 1. |
|
Line 85: especially transboundary cooperation (Poland-Ukraine-Slovakia?) |
Yes, the reviewer is right. We were thinking in general about legally formalized cross-border cooperation: Poland-Ukraine-Slovakia, Poland-Belarus, Poland-Bohemia. This phrase was added in line 89. |
|
Line 362, 363: According to the Supreme Audit Office (NIK) report, the cooperation between all Polish parks within the biosphere reserve and the cross-border cooperation with parks on the Slovak and Ukrainian sides had a more formal than practical dimension. What is situation about the cross-border cooperation in war conditions in Ukraine? No possibility for cooperation? |
We thank the reviewer for drawing attention to this aspect. Currently, under the conditions of the war in Ukraine, there are no opportunities for cooperation, due to the focus on other priorities. |
|
Line 560: and attitudes towards the economic benefits derived from nature (Mammadova et al., 2022). What are economic benefits (an existence of some ores and critical minerals? Tourismus?) |
(…) and attitudes towards the economic benefits derived from nature, that is geo- and ecotourism and tourism-related services, thanks to unique natural sites with geological objects, cultural landscapes and historical sites (Mammadova et al., 2022). |
|
Line 662, 663, 664: It is necessary, in cooperation with scientific, social, and local government communities involved in local spatial policies, to develop an action program for nature reserves in Poland ( in which period? For the first 5 years? Or until 2040?) |
This should be a strategic program, until 2040, and shorter operational programs, covering sequences of 5 years each. |
Thank you very much, Authors.
Round 2
Reviewer 3 Report
The second review of the manuscript
“Mismanagement of biosphere reserves in Poland – problems of the Polish law and nature legacy”
Sustainability-2605758
Dear Authors,
Thank you for taking into consideration my recommendations. I believe, that they have helped to improve the quality of the manuscript. I recommend the manuscript for publishing.
Two additional minor comments:
1. Lines 268-270
“There is currently no coherent spatial management in Poland. This excludes the possibility of presenting even a simplified graphical representation of the existing exact relationships between individual planning documents. There is currently no coherent spatial management in Poland. This is mainly due to.(…)” – repetition.
2. Figure 1 – Kampinos instead of “Kapminos” (2x).
Date of the Review: 2023-10-18
Author Response
Lines 268-270 - corrected; double line deleted
"Kapminos" - corrected "Kampinos" (what a shame!). Sometimes minor faults (like errors in names) go unnoticed by authors.
Thank you for your detailed review!